# A Novel Approach Based on a Weighted Interactive Network to Predict Associations of MiRNAs and Diseases

**DOI:** 10.3390/ijms20010110

**Published:** 2018-12-28

**Authors:** Haochen Zhao, Linai Kuang, Xiang Feng, Quan Zou, Lei Wang

**Affiliations:** 1College of Computer Engineering & Applied Mathematics, Changsha University, Changsha 410001, China; 2Key Laboratory of Hunan Province for Internet of Things and Information Security, Xiangtan University, Xiangtan 411105, China; zhaohc940702@163.com (H.Z.); kuanglinai@xtu.edu.cn (L.K.); fengxiang@xtu.edu.cn (X.F.); 3Institute of Fundamental and Frontier Sciences, University of Electronic Science and Technology of China, Chengdu 610000, China; zouquan@nclab.net; 4School of Computer Science and Technology, Tianjin University, Tianjin 300000, China

**Keywords:** microRNA, diseases, association prediction, computational prediction model, path selection

## Abstract

Accumulating evidence progressively indicated that microRNAs (miRNAs) play a significant role in the pathogenesis of diseases through many experimental studies; therefore, developing powerful computational models to identify potential human miRNA–disease associations is vital for an understanding of the disease etiology and pathogenesis. In this paper, a weighted interactive network was firstly constructed by combining known miRNA–disease associations, as well as the integrated similarity between diseases and the integrated similarity between miRNAs. Then, a new computational method implementing the newly weighted interactive network was developed for discovering potential miRNA–disease associations (WINMDA) by integrating the *T* most similar neighbors and the shortest path algorithm. Simulation results show that WINMDA can achieve reliable area under the receiver operating characteristics (ROC) curve (AUC) results of 0.9183 ± 0.0007 in 5-fold cross-validation, 0.9200 ± 0.0004 in 10-fold cross-validation, 0.9243 in global leave-one-out cross-validation (LOOCV), and 0.8856 in local LOOCV. Furthermore, case studies of colon neoplasms, gastric neoplasms, and prostate neoplasms based on the Human microRNA Disease Database (HMDD) database were implemented, for which 94% (colon neoplasms), 96% (gastric neoplasms), and 96% (prostate neoplasms) of the top 50 predicting miRNAs were confirmed by recent experimental reports, which also demonstrates that WINMDA can effectively uncover potential miRNA–disease associations.

## 1. Introduction

Recently, increasing studies indicated that non-coding RNAs (ncRNAs) play an extensive and important role in many biological processes such as cell differentiation, ontogenesis, and disease development [1,2,3]. In particular, microRNAs (miRNAs), a class of small ncRNAs with a length of 20–25 nucleotides, was proven to be closely related to the occurrence of many diseases that are seriously harmful to human health [4,5]; they are able to regulate many functions of eukaryotic cells and affect various behaviors such as gene expression, cell-cycle regulation, and individual development [6]. For example, miR-126 was demonstrated to be associated with clear cell human renal cell carcinoma [7], while miR-34a-5p was proven to have a critical impact on ovarian cancer (OC) cell lines via interacting with nuclear paraspeckle assembly transcript 1 (NEAT1) [8]. MicroRNA expression microarray analysis showed that miR-145 and miR-1 expression is significantly downregulated in colon cancer tissues [9], and that miR-424 and miR-381 play important roles in tumor regulation, expression, and even treatment [10]. Therefore, it is necessary to study the association between miRNAs and diseases in depth and explore the potential relationship between miRNA and some human diseases [11,12].

The identification of potential miRNA–disease associations can not only play an important role in the diagnosis, treatment, and prevention of disease, but also effectively addresses the high cost and long-term shortcomings of traditional biological experiments [13,14]. Up to now, various miRNA-related heterogeneous biological databases were established and were extended to various fields of miRNA-related research, such as miRBase [15], Database of Differentially Expressed miRNAs in Human Cancers (dbDEMC) [16], Human microRNA Disease Database (HMDD) [17], miR2Disease [18], etc. Based on these datasets, different computational prediction methods were developed to predict potential miRNA–disease associations [19,20,21,22,23,24]. For example, in 2012, Chen et al. developed a prediction method named random walk with restart for miRNA–disease association (RWRMDA) for inferring potential miRNA–disease associations through combining the semantic similarity of the disease and the functional similarity of miRNA [25]. In 2014, Chen et al. further developed a computational model named regularized least squares for miRNA–disease association (RLSMDA) based on semi-supervised machine learning, which can uncover related miRNAs for each disease without any known related miRNAs [26]. In 2016, under the basic assumption that functional similar miRNAs tend to interact with similar diseases, Chen et al. proposed a prediction model named within and between Score for miRNA–disease association (WBSMDA) by combining the within score and between score from the opinion of diseases and miRNAs to calculate the score and predict potential miRNA–disease associations [27]. Furthermore, Zou et al. also developed a machine learning based method named CATAPULT to uncover relationships between microRNAs and diseases based on social network analysis methods [28]. Zhu et al. developed a novel path-based method named PBMDA to predict the relationships between miRNA and disease by integrating different types of heterogeneous biological datasets and constructing three interlinked sub-graphs [29]. In 2017, Chen et al. proposed LRSSLMDA where novel miRNA–disease associations were predicted using sparse subspace learning to map high-dimensional miRNA/disease spaces into a lower-dimensional subspace [30]. In LRSSLMDA, feature extraction was performed on the integrated similarity to form the statistical profile and the graph theoretical profile. In 2018, Chen et al. developed a novel computational model named MDHGI to predict potential miRNA–disease associations using a sparse learning method to decompose the original adjacency matrix and combing the miRNA functional similarities network, the disease semantic similarities network, the Gaussian interaction profile kernel similarities network, and the new adjacency matrix into a heterogeneous graph [31]. Zhao and Wang presented a distance correlation set-based prediction method named DCSMDA for predicting latent miRNA–disease associations by constructing the disease/long non-coding RNA (lncRNA)/miRNA interactive network [32]. Moreover, Chen et al. developed an extreme gradient-boosting machine with a decision tree named EGBMMDA, seeking potential miRNA–disease associations using vector-covered statistical measures, graph theoretical measures, and matrix factorization results for miRNAs and diseases [33]. Chen et al. proposed a rating-integrated bipartite network recommendation-based prediction method named BNPMDA for predicting potential miRNA–disease association using agglomerative hierarchical clustering [34].

In most of the abovementioned computational models, researchers constructed an adjacency matrix A to represent relationships between miRNA and disease. Specifically, if the association term between miRNA *m_i_* and disease *d_j_* is recorded in the database, the matrix A(i,j) = 1; otherwise it is equal to zero. However, since the number of known miRNA–disease associations existing in these well-known databases is very limited, this results in a sparse matrix; thus, in order to improve the accuracy of our prediction model, the information about the diseases and miRNAs was adopted to construct a weighted interactive network in this paper. Moreover, on the basis of premises that functionally similar miRNAs may regulate similar diseases and similar diseases tend to associate with functionally similar miRNAs, a novel prediction model based on the newly constructed weighted interactive network was developed for miRNA–disease association inference (WINMDA). Comparing several state-of-the-art computational models, the strong point of WINMDA lies in the construction of a weighted interactive network and the introduction of the shortest path between nodes in the weighted interactive network. This also means that WINMDA proposes an idea for improving the sparseness of the adjacency matrix A and does not need negative samples to predict potential miRNA–disease associations simultaneously. All prediction results of potential miRNA–disease associations are shown in the Appendix A; researchers could use these data to guide biological experiments in the future. In addition, the performance of WINMDA was evaluated by cross-validation and case studies of colon neoplasms, gastric neoplasms, and prostate neoplasms. Our simulation results show that WINMDA can achieve reliable area under the receiver operating characteristics (ROC) curve (AUC) results of 0.9183 ± 0.0007, 0.9200 ± 0.0004, 0.9243, and 0.8856 in terms of 5-fold cross-validation, 10-fold cross-validation, global leave-one-out cross-validation (LOOCV), and local LOOCV, respectively. Additionally, 94% (colon neoplasms), 96% (gastric neoplasms), and 96% (prostate neoplasms) of the top 50 predicting miRNAs were confirmed by dbDEMC [16], miR2Disease [18], and recently published experimental studies. These results also demonstrate that WINMDA can effectively predict potential miRNA–disease associations.

## 2. Results and Case Studies

In this section, we evaluated the predictive performance of WINMDA through the following experiments: we firstly compared WINMDA with four state-of-the-art methods, namely BNPMDA [34], PBMDA [29], WBSMDA [27], and RLSMDA [26] in the framework of LOOCV. Then, the process of five-fold cross-validation was repeated 50 times for our method to evaluate the prediction performance of WINMDA. Thirdly, the influence of given parameters *T* and *w* on the prediction model was analyzed. Moreover, several case studies were performed to validate the feasibility of our method. Finally, experimental results regarding the top 50 predicted associations between miRNAs and four important neoplasms were listed, and we implemented the performance comparisons between WINMDA and four state-of-the-art methods through observing the rankings of six important disease-related miRNAs in the case studies.

### 2.1. Comparison with Existing State-of-the-Art Methods

We evaluated the performance of WINMDA by observing it along with four state-of-the-art methods to predict the accuracy of potential miRNA–disease associations using global and local LOOCV. In the global LOOCV, each known disease–miRNA association was alternately used as a test sample, and other known miRNA–disease associations were considered as a training set, while all unknown disease–miRNA associations in HMDD were considered as a candidate set. However, in local LOOCV, for each given disease *d*, each known miRNA related to *d* was utilized as a test sample, while all other known miRNAs related to *d* were used as training samples, and all other unknown miRNAs related to *d* were considered as candidates. Hence, through comparing the scores obtained from the test sample with other candidate associations, we could evaluate how well this association was ranked in the candidate set; if the predicted ranking of the test sample was higher than the preset threshold, then the sample was successfully predicted by the computational model. In other words, let TP be true positive, TN be true negative, FN be false negative, and FP be false positive; then, under different threshold settings, the corresponding true positive rate (TPR; sensitivity) and false positive rate (FPR; specificity) can be obtained as follows:FPR = FP/(FP + TN)(1)
TPR = TP/(TP + FN)(2)

Here, sensitivity means that the percentage of the test samples with predicted ranks was higher than the given threshold, whereas specificity was computed as the percentage of negative samples with predicted ranks lower than the given threshold. Obviously, after obtaining different TPR and FPR pairs under different thresholds, the final ROC curve (in which, FPR is the horizontal axis of the coordinate system and TPR is the longitudinal axis of the coordinate system) could be plotted by connecting these pairs. Finally, the AUC could be obtained to represent the specific prediction performance of the computational models. Furthermore, it is obvious that the larger the AUC value is, the more likely the current classification algorithm is to place a positive sample in front of the negative sample, such that it can be better classified. Next, for the global LOOCV and local LOOCV, we compared WINMDA with four state-of-the-art computational methods, namely BNPMDA, PBMDA, WBSMDA, and RLSMDA; the simulation results are shown in Figure 1 and Figure 2, respectively. From the two figures, it is easy to see that WINMDA can achieve reliable AUCs of 0.9243 and 0.8856 for global LOOCV and local LOOCV, respectively, when *T* = 16 and *w* = 0.6, which are much higher than the AUCs of 0.9169 and 0.8523 achieved by PBMDA, 0.9082 and 0.8571 achieved by BNPMDA, 0.8030 and 0.8390 achieved by WBSMDA, and 0.8426 and 0.7169 achieved by RLSMDA. It is obvious that our newly proposed method WINMDA is superior to these four traditional computational models in global and local LOOCV; therefore, it can be used as an important tool for discovering potential miRNA–disease associations.

In addition, considering the potential bias of random sample partitioning for performance assessment, we divide the known miRNA-disease associations by 50 times, and the AUCs were obtained in the similar way as global LOOCV. As a result, WINMDA achieved the prediction performance with average AUCs of 0.9183 and 0.9200 with standard deviation of 0.0007 and 0.0004 when using the 5-Fold and 10-Flod cross validation (Table 1).

### 2.2. Evaluation of the Effects of Parameters

There are two kinds of important parameters existing in our newly proposed model of WINMDA, as illustrated in Equation (22), one is *w* and the other is *T*.

#### 2.2.1. Effects of Parameter *T*

From Equations (20), (21), and (22), it is easy to know that parameter *T* will have important effects on the accuracy of our prediction model WINMDA. For instance, if the value of *T* is too large, then lots of noise data will be included, which will reduce the predictive performance of WINMDA. Alternatively, if the value of *T* is too small, then the useful associations may not be sufficient for accurate estimation of potential associations between some specific diseases and miRNAs. Hence, in order to evaluate the effects of parameter *T*, we set the value of *T* ranging from 1–20 during the implementation of WINMDA, and the simulation results are shown in Table 2. From Table 2, it is easy to see that the AUCs achieved by WINMDA varied with the different values of *T*. Specifically, the prediction performance of WINMDA increased upon increasing of the value of *T*, while *T* varied from 1 to 16, which indicates that the number of useful neighbors is positively related to the prediction performance of our model. Meanwhile, it is easy to find that the AUC will decline when *T* > 16, which indicates that an excess of noise data will markedly interfere with our prediction model WINMDA. Therefore, it was determined as best to set *T* to 16 for WINMDA in this paper.

#### 2.2.2. Effects of Parameter w

In this section, to investigate the effects of parameter *w* on the prediction performance of WINMDA, we set *w* to different values ranging from 0–1, while implementing WINMDA under LOOCV, and the results are shown in the Table 3. It is obvious that the variation of the value of *w* has an important influence on the performance of our prediction model WINMDA. Specifically, from Table 3, it is obvious that WINMDA can achieve the maximum AUC value when *w* is set to 0.6. Hence, we set *w* to 0.6 in this paper.

### 2.3. Case Studies

Recently, increasing evidence demonstrated that miRNAs play an extensive and important role in the physiological processes of the body [35]. In addition, in developed countries such as the United States and throughout Europe, cancer is the second leading cause of human death, while it ranks second or third in developing countries [36]. Therefore, in order to further evaluate the accuracy of WINMDA in predicting potential disease–miRNA associations, we chose three kinds of cancers, i.e., colon neoplasms, gastric neoplasms, and prostate neoplasms, as case studies for WINMDA, and the prediction results were verified by recently published experimental studies and two databases, namely miR2Disease and dbDEMC. During the simulation, for each kind of cancer, all known related miRNAs were considered as seed miRNAs, and the other miRNAs were considered as candidate miRNAs. In addition, all candidate miRNAs associated with colon neoplasms, gastric neoplasms, and prostate neoplasms were ranked in descending order according to our prediction results, as illustrated in Table 4, Table 5 and Table 6, respectively.

Recently, colon cancer (colon neoplasms) ranks third among the most common female cancers and second among the most common male cancers in the world [37]. Each year, more than one million people died from colon cancer [38]. The incidence rates vary widely around the world, depending on lifestyle, environment, and heredity [39]. Recent studies reported that miRNAs are closely related to the diagnosis, prognosis, and chemo-sensitivity of colon cancer, which indicates that miRNAs can be used as a marker for the early diagnosis of colon cancer and as a guideline for various stages of colon cancer [40]. Hence, case studies on colon cancer-related miRNAs were implemented to further verify the predictive ability of WINMDA and, as a result, 10 of the top 10 and 47 of the top 50 candidate miRNAs were shown to be associated with colon neoplasms by miR2Disease, dbDEMC, and other known experimental studies (Table 4). For example, some researchers confirmed that the overexpression of miR-143 (ranked first in the WINMDA forecast list) reduces cell proliferation and migration of mutant *KRAS* HCT116 colon cancer cells [41]. Additionally, experimental studies also found that miR-20a is a member of the miR-17 miRNA family, which is part of the regulatory machinery that defines the pro-tumorigenic differentiation of stromal fibroblasts. In stromal fibroblasts, miR-20a (ranked second in the WINMDA forecast list) can modulate chemokine C–X–C ligand 8 (CXCL8) function, thereby influencing tumor latency [42]. Moreover, some researchers found that the rs35301225 polymorphism in miR-34a (ranked third in the WINMDA forecast list) is involved in the development of human colon cancer via downregulation of tumor-promoting gene *E2F1* as a tumor suppressor, and the C/A single-nucleotide polymorphism of miR-34a promotes colon cancer cell proliferation via upregulating *E2F1* [43].

In recent years, it was reported that gastric cancer (gastric neoplasms) is one of the most common malignant tumors of the digestive tract in the world, and Japan, South Korea, and China are high-risk areas for gastric cancer [44]. Therefore, it is necessary to explore the mechanism of miRNA in the development of gastric cancer and provide a basis for the early diagnosis of gastric cancer. We used potential gastric cancer-associated miRNAs as a case study to further illustrate the predictive power of WINMDA in this section. As a result, 10 of the top 10 and 48 of the top 50 potential gastric cancer-related miRNAs were validated by miR2Disease, dbDEMC, and other known experimental studies (Table 5). For example, gene *UHRF1* plays a significant role in the development of gastric cancer. Furthermore, Zhou et al. identified and verified miR-146b (ranked first in our prediction list) and miR-146a as direct upstream regulators of *UHRF1* in gastric cancer metastasis. [45]. In addition, according to the target genes of miR-143 (ranked seventh in our prediction list), *IGF1R* and *BCL2*, which are related to cisplatin resistance, we can regulate the resistance of human gastric cancer cells to cisplatin via differential expression of *IGF1R* and *BCL2* in gastric cancer tissues and cell lines [46].

Prostate cancer (prostate neoplasms) is the third most common type of cancer. In 2012, the incidence rate of prostate cancer in the neoplasm registration area in China was 99.2%, which ranked sixth in the incidence of male malignant tumors [47]. However, early patients with prostate tumors have only subtle symptoms that make it difficult to detect cancer at an early stage [48]. Increasing studies confirmed that some miRNAs are related to prostate neoplasms. Therefore, case studies about prostate cancer-related miRNAs were implemented to further verify the predictive ability of WINMDA in this section. As a result, nine of the top 10 and 48 of the top 50 predicted prostate cancer-related miRNAs were validated by miR2Disease, dbDEMC, and other known experimental studies (Table 6). For example, Chu et al. selected single-nucleotide polymorphisms (SNPs) in the 1000 bp upstream from the transcription start site of hsa-miR-143 (ranked first in our prediction list) precursor in the dbSNP database with the condition that MAF > 0.05 in the Chinese population and finally identified that rs4705342 T > C was associated with the risk of prostate cancer, and that the C allele had a protective effect [49]. Wang et al. explored the effects of miR-182 (ranked second in our prediction list) on the growth, migration, and apoptosis of prostate cancer cells using qRT-PCR analysis. Moreover, they found that miR-182 plays an important role in prostate cancer, which enhances HIF1α signaling by targeting PHD2 and FIH1 in prostate cancer [50]. Furthermore, Taddei et al. confirmed that hsa-miR-210 (ranked fourth in our prediction list) overexpression increased senescence-associated features in young fibroblasts and converted them into cancer-associated fibroblast-like cells. These senescent fibroblasts can induce epithelial–mesenchymal transition in prostate cancer cells, support tumor angiogenesis, and recruit endothelial precursor cells, thus contributing to cancer progression [51].

In order to further illustrate the high efficiency of our method, we compared the performances of PBMDA, WBSMDA, LRLSMDA, and our model WINMDA by counting the top 50 disease-associated miRNAs in the predicted results and observing how many disease-related miRNAs were identified by miR2Disease, dbDEMC, and recent biological experimental studies (Table 7) in the case studies of six important diseases. As a result, from Table 7, it is easy to see that WINMDA is more effective than other methods in general. In addition, as a global computational model, WINMDA can not only achieve reliable prediction performances, but also simultaneously predict all potential associations between the diseases and miRNAs in HMDD, which means that potential associations with high predicted values obtained by WINMDA can be used preferentially for biological experiment verification and public release. Hence, we may easily reach a conclusion that our newly proposed model WINMDA is of great value in predicting potential miRNA–disease associations.

## 3. Discussion

Increasing studies based on biological experiments indicated that miRNAs are closely related to the occurrence of many diseases that are seriously harmful to human health, and the identification of potential miRNA–disease associations can not only play an important role in the diagnosis, treatment, and prevention of disease, but also effectively addresses the high cost and long-term shortcomings of traditional biological experiments. In this article, we developed a novel prediction model called WINMDA to predict potential relationships between miRNAs and diseases based on premises that functionally similar miRNAs may regulate similar diseases and similar diseases tend to associate with functionally similar miRNAs. In WINMDA, we firstly integrated disease semantic similarity, Gaussian interaction profile kernel similarity, and miRNA function similarity, and then constructed a weighted interactive network for potential miRNA–disease prediction. The important difference between WINMDA and previous state-of-the-art prediction models is that the problem of limited known miRNA–disease associations was considered in WINMDA and the shortest paths in the weighted interactive network were adopted to solve the problem. Moreover, we evaluated the predictive performance of WINMDA through LOOCV (including global LOOCV and local LOOCV), *k*-fold cross-validation, and several case studies. Experimental results show that WINMDA can effectively uncover potential disease–miRNA candidates, which means that it can be used as a reliable and accurate calculation model for finding potential miRNA–disease associations.

Although WINMDA achieved effective performance in predicting candidate relationships between miRNAs and diseases, there are still some existing limitations that can be improved in the future. Firstly, the parameters *T* and *w* play important roles in WINMDA, and the selection of suitable values for *T* and *w* are critical problems that shall be addressed in future studies. Secondly, the assigned weight may not be accurate enough, as it was on the basis of premises that functionally similar miRNAs may regulate similar diseases and similar diseases tend to associate with functionally similar miRNAs. Finally, a weighted interactive network was constructed in WINMDA based on the disease similarity, miRNA similarity, and known miRNA–disease associations. The performance of WINMDA will be further improved considering more databases storing other information about diseases, miRNAs, and miRNA–disease associations. 

## 4. Materials and Methods 

### 4.1. Construction of the miRNA–Disease Interactive Network

In order to construct the miRNA–disease interactive network, we firstly downloaded known miRNA–disease associations from the HMDD database on 14 July 2018. After eliminating duplicate values, erroneous data, and disorganized data, human miRNA–disease associations were downloaded from the HMDD database, which includes 5430 experimentally validated human miRNA–disease associations involving 495 miRNAs and 383 diseases. (Appendix A) have been collected. Let *D* represent the number of different disease items and *M* represent the number of different miRNA items in the HMDD database, respectively; let *S_D_* = {*d*_1_*, d*_2_*, ..., d_D_*} represent the set of these *D* different diseases, and *S_M_*= {*m_D+_*_1_*, m_D+_*_2_*, ..., m_D+M_*} represent the set of these *M* different miRNAs. Then, we can construct an miRNA–disease interactive network *G =* (*V, E*), where *V* = *S_D_*∪*S_M_* = {*d_1_, d_2,_ ..., d_D_, m_D+1_, m_D+2_..., m_D+M_*} is the set of vertices, *E* is the edge set of *G*, and ∀ *d_i_* ∈ *S_D_*, *m_j_* ∈ *S_M_*. There is an edge between *d_i_* and *m_j_* in *E* if and only if there is an association between *m_j_* and *d_i_* in the database of HMDD. Thereafter, based on the newly constructed miRNA–disease interactive network *G*, for any given *d_i_*
∈
*S_D_* and *m_j_* ∈ *S_M_*, we can obtain a *D × M* dimensional matrix *DMM* as follows:(3)$$DMM\left(i,j\right)=\{\begin{array}{cc} 1\; & if\;d_{i}\;is\;related\;to\;m_{j}\;in\;HMDD\\ 0 & otherwise \end{array}$$

### 4.2. Calculation of the Disease Semantic Similarity

For any two diseases *d_i_* and *d_j_* that belong to *S_D_*, the semantic similarity between *d_i_* and *d_j_* was calculated according to the following steps:

**Step 1**: Firstly, we collected the Medical Subject Headings (MeSH) descriptors of *d_i_* and *d_j_* from the National Library of Medicine (http://www.nlm.nih.gov/).

**Step 2**: Secondly, we constructed direct acyclic graphs (DAGs) corresponding to *d_i_* and *d_j_* separately and, as illustrated in the Figure 3, for any given disease *H*, its DAG can be represented as DAG(*H*) = (*N*(*H*), *E*(*H*)), where *N*(*H*) is the node set and *E*(*H*) is the edge set. Moreover, in DAG(*H*), each node corresponds to a different disease MeSH descriptor, and all the MeSH descriptors are connected by a direct edge from a more general term (called a parent node) to a more specific term (called a child node). Furthermore, *N*(*H*) consists of the node *H* itself and its ancestor nodes; *E*(*H*) consists of corresponding direct edges from a parent node to a child node, and each edge in *E*(*H*) represents the relationship between these two nodes connected by it. 

**Step 3**: Thirdly, based on the newly constructed DAG(*H*), let *d* be an ancestor node of *H* in DAG(*H*); then, we defined the contribution of an ancestor node *d* to the semantic value of the disease *H* and the contribution of the semantic value of disease *H* itself as follows:(4)$$\{\begin{array}{cc} D_{H}\left(d\right)=1 & if\;d=H;\\ D_{H}\left(d\right)=\sum \left\{\alpha \;\times \beta \times D_{H}\left(d^{*}\right)|d^{*}\in children\;of\;d\right\}\; & if\;d\neq H. \end{array}$$

Here, the parameter α is a semantic contribution attenuation factor with a value between zero and one, and its value was set to 0.5 in this paper according to previous state-of-the-art methods [52,53]. The parameter β is the number of addresses or codes included in the node $d^{*}$, which indicates the weight of the contribution of disease $d^{*}\;$for *H* in DAG(*H*). 

Obviously, according to Equation (2), it is easy to know that an ancestor node *d* with a larger number of child nodes in DAG(*H*) will make a more significant contribution to the semantic value of *H*. For instance, in DAG(BN) of Figure 3, the entry on “central nervous system neoplasms” includes two addresses or codes: C04.588.614.250 and C10.551.240; however, the entry on “brain diseases” includes only one code: C10.228.140. Thus, the contribution of “central nervous system neoplasms” to the semantic value of the “brain neoplasms” is 2 × α × 1, while the contribution of “brain diseases” to the semantic value of the “brain neoplasms” is 1 × α × 1 only.

**Step 4:** Next, based on Equation (2), we calculated the sematic value of disease *H* by accumulating the contributions of all disease terms to *H* in *DAG*(*H*) as follows:(5)$$TS\left(H\right)=\sum_{d\in S\left(H\right)}D_{H}\left(d\right)$$

For example, according to Equation (3), in DAG(BN) of Figure 3, the semantic value of the disease “brain neoplasms” can be obtained by “the contributions of ‘brain neoplasms’ to it” (= 1 × 1) + “the contributions of ‘central nervous system neoplasms’ to it” (= 2 × 0.5) + “the contributions of ‘brain diseases’ to it” (= 1 × 0.5) + “the contributions of ‘nervous system neoplasms’ to it” (= 2 × 0.5 × 0.5) + “the contributions of ‘central nervous system diseases’ to it” (= 1 × 0.5 × 0.5) + “the contributions of ‘nervous system diseases’ to it” (= 1 × 0.5 × 0.5 × 0.5 + 1 × 0.5 × 0.5 × 0.5) + “the contributions of ‘neoplasms by site’ to it” (= 1 × 0.5 × 0.5 × 0.5) + “the contributions of ‘neoplasms’ to it” (= 1 × 0.5 × 0.5 × 0.5 × 0.5) = 3.6875.

**Step 5:** Finally, we defined the semantic similarity between *d_i_* and *d_j_* as follows:(6)$$SDD\left(d_{i},d_{j}\right)=\frac{\sum_{d\in S\left(d_{i}\right)\cap S\left(d_{j}\right)}\left(D_{d_{i}}\left(d\right)+D_{d_{j}}\left(d\right)\right)}{TS\left(d_{i}\right)+TS\left(d_{j}\right)}$$

Additionally, for any two diseases *d_a_* and *d_b_* that belong to *S_D_*, if *d_a_* and *d_b_* do not have semantic similarity, we define $SDD\left(d_{a},d_{b}\right)=$ −1; then, based on Equation (4), it is obvious that we can obtain a *D* × *D* dimensional disease semantic similarity matrix *SDD* (*i*, *j*).

### 4.3. Calculation of the miRNA Functional Similarity

Considering that, in the HMDD database, one miRNA may be associated with multiple disease items and vice versa, and, according to the state-of-the-art literature [39], the functional similarity can be obtained by integrating the semantic similarity of the two groups of diseases associated with these two miRNAs, then, for any two diseases *m_i_* and *m_j_* that belong to *S_M_*, the functional similarity between *m_i_* and *m_j_* can be calculated according to the following steps:

**Step 1**: Firstly, let *dx* be any given disease, and *Dgroup*= {*dy*_1_, *dy*_2_, *dy*_3_…. *dy_r_*} be a set consisting of *r* different diseases, and then the semantic similarity between *dx* and *Dgroup* can be calculated as follows:(7)$$SS\left(dx,Dgroup\right)=\begin{array}{c} \max\\ 1\leq i\leq r \end{array}\left(SDD\left(dx,dy_{i}\right)\right)$$

For example, let *d_a_*, *d_b_*, and *d_c_* be three kinds of diseases, *Dgroup* = {*d_b_*, *d_c_*}, *SDD* (*d_a_*, *d_b_*) = 0.7, and *SDD* (*d_a_*, *d_c_*) = 0.8; then, the semantic similarity between *d_a_* and *Dgroup* is *SS* (*d_a_*, *Dgroup*) = *max* {*SDD* (*d_a_*, *d_b_*), *SDD* (*d_a_*, *d_c_*)} = 0.8.

**Step 2**: Secondly, let *Dgroup_i_* and *Dgroup_j_* be the sets of diseases associated with *m_i_* and *m_j_*, respectively; supposing that there are *N* and *M* different diseases in *Dgroup_i_* and *Dgroup_j_*, then we calculated the functional similarity between *m_i_* and *m_j_* as follows:(8)$$SMM\left(m_{i},m_{j}\right)=\frac{\sum_{1\leq i\leq M}SS\left(d_{i},\;Dgroup_{i}\right)+\sum_{1\leq j\leq N}SS\left(d_{j},\;Dgroup_{j}\right)}{M+N},$$where *d_i_*
∈
*Dgroup_j_* and *d_j_*
∈
*Dgroup_i_*. For example, let *Dgroup*_1_ = {*X*, *Y*}, *Dgroup*_2_ = {*X*, *Z*}, supposing that *SDD*(*X*, *Y*) = 0.6, *SDD*(*X*, *Z*) = 0.7, and *SDD*(*Y*, *Z*) = 0.5, then *MM*(*m_1_*,*m_2_*) can be obtained as follows:
$$\begin{array}{c} SMM\left(m_{1},m_{2}\right)=\frac{SS\left(X,Dgroup_{1}\right)+SS\left(Y,Dgroup_{2}\right)+SS\left(X,Dgroup_{1}\right)+SS\left(Z,Dgroup_{2}\right)}{2+2}\\ =\frac{SDD\left(X,X\right)+SDD\left(Y,X\right)+SDD\left(X,X\right)+SDD\left(Z,X\right)}{4}=\frac{1+0.6+1+0.7}{4}=0.825. \end{array}$$

Additionally, for any two miRNAs *m_a_* and *m_b_* that belong to *S_M_*, if *m_a_* and *m_b_* do not have semantic similarity, we define $SMM\left(m_{a},m_{b}\right)=\;$−1; then, based on Equation (6), it is obvious that we can obtain an *M × M* dimensional miRNA functional similarity matrix *SMM* (*i*, *j*).

### 4.4. Disease Gaussian Interaction Profile Kernel Similarity Measurement

On the basis of premises that functionally similar miRNAs may regulate similar diseases and similar diseases tend to associate with functionally similar miRNAs, let *DLP*(*d_i_*) represent the *i*-th row in the matrix *DMM*; then, for any two diseases *d_i_* and *d_j_* that belong to *S_D_*, we can calculate the Gaussian interaction profile kernel similarity between them as follows:(9)$$DGS\left(d_{i},d_{j}\right)=\exp\left(-\frac{D\times {\left|\left|DLP\left(d_{i}\right)-DLP\left(d_{j}\right)\right|\right|}^{2}}{\sum_{i=1}^{D}{\left|\left|DLP\left(d_{i}\right)\right|\right|}^{2}}\right)$$

Additionally, based on previous work [54], we can further improve the disease Gaussian interaction profile kernel similarity using a logistic function as follows:(10)$$FDGS\left(i,j\right)=\frac{1}{1+e^{-15*\mathrm{DGS}\left(d_{i},d_{j}\right)+\log\left(9999\right)}}$$

### 4.5. MicroRNA Gaussian Interaction Profile Kernel Similarity Measurement

On the basis of premises that functionally similar miRNAs may regulate similar diseases and similar diseases tend to associate with functionally similar miRNAs, let *MLP*(*m_i_*) represent the *i-*th column in the matrix *DMM*; then, for any two miRNAs *m_i_* and *m_j_* that belong to *S_M_*, we can calculate the Gaussian interaction profile kernel similarity between them as follows:(11)$$FMGS\left(i,j\right)=\exp\left(-\frac{M*{\left|\left|MLP\left(m_{i}\right)-MLP\left(m_{j}\right)\right|\right|}^{2}}{\sum_{i=1}^{M}{\left|\left|MLP\left(m_{i}\right)\right|\right|}^{2}}\right).$$

### 4.6. Calculation of the Integrated Similarity

Based on Equations (6) and (10), the disease integrated similarity matrix *FDD* can be calculated based on the disease semantic similarity matrix (*SDD*) and the disease Gaussian interaction profile kernel similarity matrix (*FDG*S) as follows:(12)$$FDD\left(i,j\right)=\{\begin{array}{cc} SDD\left(i,j\right)\; & if\;SDD\left(i,j\right)\geq 0,\\ FDGS\left(i,j\right) & otherwise. \end{array}$$

Similarly, based on Equations (8) and (11), the miRNA integrated similarity matrix *FMM* can be calculated based on the miRNA functional similarity matrix (*SMM*) and the miRNA Gaussian interaction profile kernel similarity matrix (*FMGS*) as follows:(13)$$FMM\left(i,j\right)=\{\begin{array}{cc} SMM\left(i,j\right)\; & if\;SMM\left(i,j\right)\geq 0,\\ FMGS\left(i,j\right) & otherwise. \end{array}$$

### 4.7. Construction of the Weighted Interactive Network

For any given miRNA *m_i_*
∈
*S_M_*, we define the miRNA *m_x_*
∈
*S_M_* as the most related miRNA to *m_i_*, if *m_x_* satisfies the following:(14)$$FMM\left(m_{x},m_{i}\right)=\begin{array}{c} max\\ 1\leq l\leq M \end{array}\left(FMM\left(m_{l},m_{i}\right)\right),\;where\;m_{x}\neq m_{i}.$$

Thereafter, as illustrated in Figure 4, we can construct the weighted interactive network according to the following four steps:

**Step1**: Firstly, for any given disease *d_i_*∈ *S_D_*, we define the miRNA *m_j_* as a potential miRNA to *d_i_* if and only if *m_j_* satisfies *DMM*(*i,j*) = 0; otherwise, we define the miRNA *m_j_* as a known miRNA to *d_i_*. Hence, according to premises that functionally similar miRNAs may regulate similar diseases and similar diseases tend to associate with functionally similar miRNAs, it is reasonable to assume that the miRNA *m_j_* is related to *d_i_* if *m_j_* is a potential miRNA to *d_i_*, *m_x_* is a most related miRNA to *m_j_*, and *m_x_* is also a known miRNA to *d_i_* at the same time. Thereafter, based on this assumption, for any given disease *d_i_*
∈ *S_D_* and any given miRNA *m_j_*
∈ *S_M_*, we can define the weight between *d_i_* and *m_j_* as follows:
(15)$$DMW\left(i,j\right)=\{\begin{array}{cc} \frac{1}{\exp\left(\underset{x\in MRM\left(m_{j}\right)}{\max}SMM\left(j,x\right)\right)}\; & if\;DMM\left(i,j\right)=0\;and\;DMM\left(i,x\right)\neq 0,\\ \frac{1}{\exp\left(DMM\left(i,j\right)\right)} & if\;DMM\left(i,j\right)\neq 0,\\ 0 & otherwise. \end{array}$$

**Step 2**: Secondly, according to Equation (10), for any two given diseases *d_i_* and *d_j_* that belong to *S_D_*, we define the weight between *d_i_* and *d_j_* as follows:(16)$$DW\left(i,j\right)=\frac{1}{\exp\left(FDD\left(i,j\right)\right)}.$$

From Equation (14), it is easy to know that the higher the semantic similarity between *d_i_* and *d_j_* is, the smaller the weight between *d_i_* and *d_j_* will be.

**Step 3**: Similarly, according to Equation (11), for any two given miRNAs *m_i_* and *m_j_* that belong to *S_M_*, we define the weight between *m_i_* and *m_j_* as follows:(17)$$MW\left(i,j\right)=\frac{1}{\exp\left(FMM\left(i,j\right)\right)}$$

From Equation (15), it is also easy to know that the higher the functionally similarity between *m_i_* and *m_j_* is, the smaller the weight between *m_i_* and *m_j_* will be.

Thereafter, based on above three steps, for *i* ∈ [1*,D* + *M*] and *j* ∈ [1*,D + M*], a weighted miRNA–disease interactive network can finally be constructed as follows:
(18)$$GFW\left(i,j\right)=\{\begin{array}{cc} DW\left(d_{i},d_{j}\right),\; & if\;i\in \left[1,D\right]\;\mathrm{and}\;j\in \left[1,D\right],\\ DMW\left(d_{i},m_{j}\right), & if\;i\in \left[1,D\right]\;\mathrm{and}\;j\in \left[D,D+M\right],\\ DMW\left(m_{i},d_{j}\right), & if\;i\in \left[D,D+M\right]\;\mathrm{and}\;j\in \left[1,D\right],\\ MW\left(m_{i},m_{j}\right), & if\;i\in \left[D,D+M\right]\;\mathrm{and}\;j\in \left[D,D+M\right]. \end{array}$$

### 4.8. Calculation of the Shortest Path Based on the Weighted Interactive Network

For any two given nodes *A* and *B* in the weighted interactive network *G*, supposing that there is a path *P* consisting of *n* hops such as *P*_0_(=*A*), *P*_1_, *P*_2_,…,*P_n_*(=*B*) from *A* to *B* in *G*, then we define the weights of path *P* as $\sum_{i=0}^{n-1}GFW\left(i,i+1\right)$. In addition, among all the paths from *A* to *B* in *G*, a path from *A* to *B* with smallest weights is called the shortest path from *A* to *B*. Thereafter, it is reasonable to assume that, for any two given nodes *A* and *B* in the weighted interactive network *G*, the smaller the weight of the shortest path from *A* to *B* is, the more related to each other the nodes *A* and *B* will be. Thus, based on this assumption, for any two given nodes *A* and *B* in the weighted interactive network *G*, we can design an algorithm for searching the shortest path from *A* to *B* in *G* as follows:

**Step 1**: Initially, we define that *S* = {*V*_0_} is a set consisting of an arbitrary node *V*_0_ in *G*, *T* is a set consisting of all nodes in *G* other than *V_0_*, and *DD* is a matrix defined as follows:(19)$$DD\left(i,j\right)=\{\begin{array}{cc} GFW(i,j)\; & if\;GFW\left(i,j\right)>0,\\ \infty & if\;GFW\left(i,j\right)=0, \end{array}$$
where *i* ∈ [1*,D*+*M*] and *j* ∈ [1*,D+M*].

**Step 2**: Next, we select a node *V_k_* from *T* randomly, if *V_k_* satisfies that *V_k_*
∉
*S* and the distance from *V_k_* to *S* is smaller than the distance from any other node other than *V_k_* in *T* to *S*. Here, we define the distance from a node *x* in *G* to a node set *V* in *G* as the smallest value of the distances between *x* and all nodes in *V*.

**Step 3**: Thereafter, if *DD* (*i, j*) > *DD* (*i, k*) + *DD* (*k, j*), then we further modify *DD*(*i,j*) to *DD*(*i, j*) = *DD*(*i, k*) + *DD*(*k, j*) in the matrix *DD*.

**Step 4**: After repeating step 2 and step 3 until all nodes in *G* are included in *S*, then it is obvious that we can transfer the matrix *DD* to a (*D* + *M*) × (*D* + *M*) dimensional shortest path matrix (*SPM*).

### 4.9. Calculation of the Shortest Path Based on the Weighted Interactive Network

Considering the fact that known miRNA–disease associations are very sparse, for a specific disease *d_i_* and a specific miRNA *m_j_*, as illustrated in Figure 5, in this section, we adopt the concept of *T* most similar neighbors to estimate the association between *d_i_* and *m_j_* according to the following steps:

**Step 1**: Firstly, for the disease *d_i_*, let *DK_i_* = {*d_i_*_1_, *d_i_*_2_, *d_i_*_3_…., *d_iT_*} be a set consisting of the first *T* nodes in *S_D_* after sorting the nodes in *S_D_* by the disease integrated similarity between them with *d_i_* in descending order, and, for the miRNA *m_j_*, let *MK_j_* = {*m_j_*_1_, *m_j_*_2_, *m_j_*_3_…., *m_jT_*} be a set consisting of the first *T* nodes in *S_M_* after sorting the nodes in *S_M_* by the miRNA integrated similarity between them with *m_j_* in descending order.

**Step 2**: Secondly, according to premises that functionally similar miRNAs may regulate similar diseases and similar diseases tend to associate with functionally similar miRNAs, we calculate the association between *d_i_* and *MK_j_* and the association between *m_j_* and *DK_i_* as follows:(20)$$SDM\left(d_{i},MK_{j}\right)=\sum_{1<q<T}SPM\left(d_{i},m_{jq}\right)$$
(21)$$SMD\left(m_{j},MD_{i}\right)=\sum_{1<q<T}SPM\left(m_{j},d_{iq}\right)$$

**Step 3**: In order to optimize the prediction results, by integrating the above two associations and the matrix *SPM*, we can obtain our final prediction results as follows:(22)$$FPR\left(i,j\right)=w\times \frac{\left(SDM\left(d_{i},MK_{j}\right)+SMD\left(m_{j},MD_{i}\right)\right)}{2\times T}+\left(1-w\right)\times SPM\left(i,j\right)$$where *w* is a weight coefficient with a value from zero to one.

## 5. Conclusions

In this article, the effective predictive performance of WINMDA was mainly due to several reasons. Firstly, the sematic disease similarity, functional miRNA similarity, and Gaussian interaction profile kernel similarity were integrated. Secondly, we proposed a new method for calculating the semantic similarity of diseases. Thirdly, we constructed a weighted interactive network-based disease similarity, miRNA similarity, and known miRNA–disease associations. Fourthly, the concept of *T* most similar neighbours was introduced. Finally, an algorithm for searching the shortest path in the weighted interactive network was introduced. Furthermore, in future work, multiple heterogeneous biological data can be collected and pre-processed to be utilized in the weighted interactive network, thus improving the performance of prediction algorithms.

## Figures and Tables

**Figure 1 ijms-20-00110-f001:**
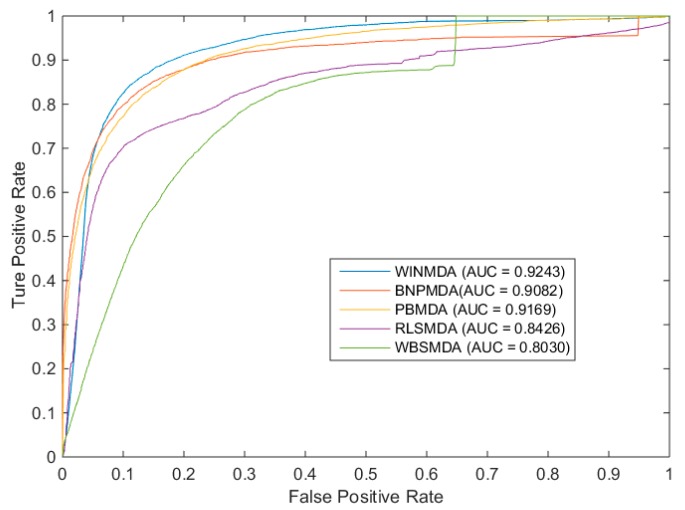
The comparison results between the weighted interactive network for miRNA–disease association inference (WINMDA) and four state-of-the-art computational models in terms of global leave-one-out cross-validation (LOOCV).

**Figure 2 ijms-20-00110-f002:**
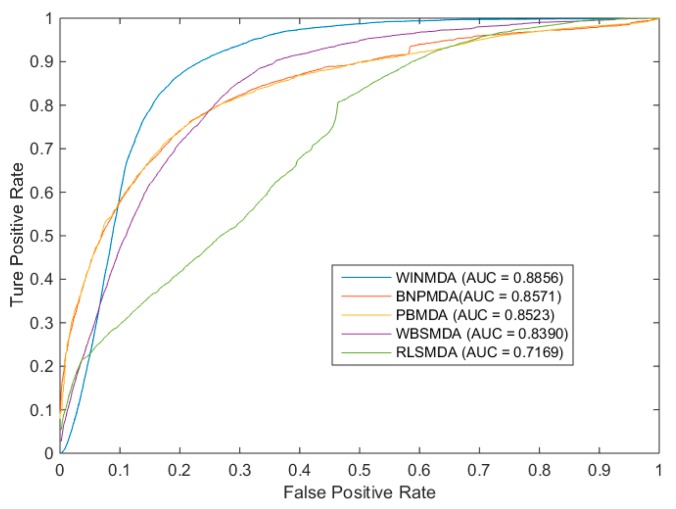
The comparison results between WINMDA and four state-of-the-art computational models in terms of local LOOCV.

**Figure 3 ijms-20-00110-f003:**
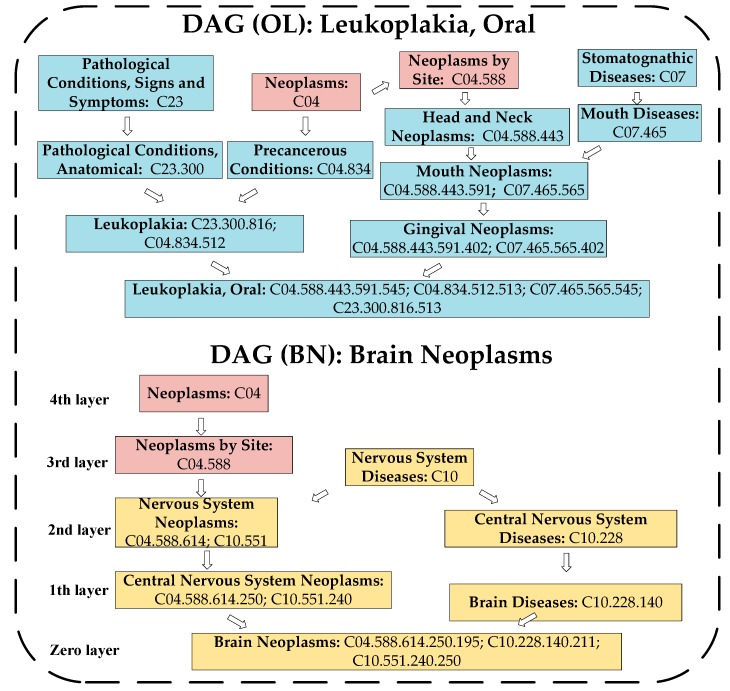
The disease directed acyclic graphs (DAGs) of leukoplakia, and oral and brain neoplasms.

**Figure 4 ijms-20-00110-f004:**
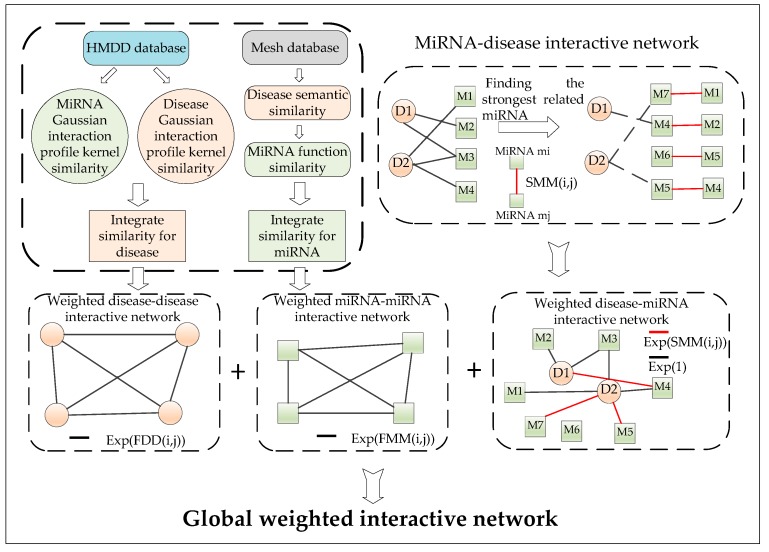
The flowchart detailing the construction of the global weighted interactive network by combining the weighted disease–disease interactive network, the weighted miRNA–miRNA interactive network, and the weighted disease–miRNA interactive network.

**Figure 5 ijms-20-00110-f005:**
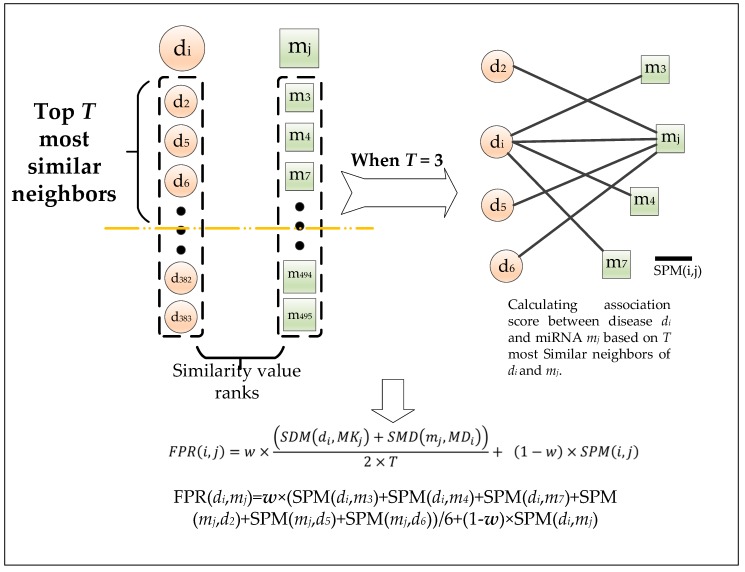
The process of predicting potential miRNA–disease associations based on the concept of *k* most similar neighbors.

**Table 1 ijms-20-00110-t001:** The average area under the receiver operating characteristics (ROC) curve (AUC) achieved by the weighted interactive network for miRNA–disease association inference (WINMDA) under the frameworks of 5-Fold cross-validation and 10-Fold cross-validation.

LOOCV	5-Fold Cross-Validation	10-Fold Cross-Validation
0.9243	0.9183 ± 0.0007	0.9200 ± 0.0004

**Table 2 ijms-20-00110-t002:** Effects of *T* on the prediction performance of WINMDA when *w* = 0.6.

*T*	AUC	*T*	AUC
1	0.9145	12	0.9241
2	0.9160	*16*	*0.9243*
5	0.9222	18	0.9242
8	0.9244	20	0.9188

**Table 3 ijms-20-00110-t003:** Effects of *w* on the prediction performance of WINMDA when *T* = 16.

*w*	AUC	*w*	AUC
0	0.9135	0.6	0.9243
0.1	0.9188	0.7	0.9239
0.2	0.9209	0.8	0.9222
0.3 0.4 0.5	0.9216 0.9235 0.9241	0.9 1	0.9189 0.9160

**Table 4 ijms-20-00110-t004:** The potential top 50 predicted microRNAs (miRNAs) related to colon neoplasms obtained by WINMDA based on known associations in the Human microRNA Disease Database (HMDD) database.

Top 1–25 miRNAs	Evidence	Top 26–50 miRNAs	Evidence
hsa-mir-143	dbDEMC and miR2Disease	hsa-let-7e	dbDEMC
hsa-mir-20a	dbDEMC and miR2Disease	hsa-mir-486	26895105
hsa-mir-34a	dbDEMC and miR2Disease	hsa-mir-133b	dbDEMC and miR2Disease
hsa-mir-210	dbDEMC	hsa-mir-200a	unconfirmed
hsa-mir-21	dbDEMC and miR2Disease	hsa-mir-141	dbDEMC and miR2Disease
hsa-mir-155	dbDEMC and miR2Disease	hsa-let-7f	dbDEMC and miR2Disease
hsa-mir-95	dbDEMC and miR2Disease	hsa-mir-29a	dbDEMC and miR2Disease
hsa-mir-146a	dbDEMC	hsa-mir-181a	dbDEMC and miR2Disease
hsa-mir-16	dbDEMC	hsa-mir-9	dbDEMC and miR2Disease
hsa-mir-125b	dbDEMC	hsa-mir-29b	dbDEMC and miR2Disease
hsa-mir-92a	unconfirmed	hsa-let-7c	dbDEMC
hsa-mir-31	dbDEMC and miR2Disease	hsa-let-7d	dbDEMC
hsa-mir-223	dbDEMC and miR2Disease	hsa-mir-196a	dbDEMC and miR2Disease
hsa-mir-221	dbDEMC and miR2Disease	hsa-let-7i	dbDEMC
hsa-mir-222	dbDEMC	hsa-mir-142	23619912
hsa-let-7a	dbDEMC and miR2Disease	hsa-mir-1	dbDEMC and miR2Disease
hsa-mir-19b	dbDEMC and miR2Disease	hsa-mir-133a	dbDEMC and miR2Disease
hsa-mir-15a	dbDEMC	hsa-mir-192	dbDEMC and miR2Disease
hsa-mir-18a	dbDEMC and miR2Disease	hsa-mir-150	26455323
hsa-mir-200b	dbDEMC	hsa-mir-203	dbDEMC and miR2Disease
hsa-mir-19a	dbDEMC and miR2Disease	hsa-mir-451a	25484364
hsa-let-7b	dbDEMC and miR2Disease	hsa-let-7g	dbDEMC and miR2Disease
hsa-mir-24	miR2Disease	hsa-mir-124	dbDEMC
hsa-mir-199a	unconfirmed	hsa-mir-224	dbDEMC and miR2Disease
hsa-mir-200c	dbDEMC and miR2Disease	hsa-mir-146b	28466779

**Table 5 ijms-20-00110-t005:** The potential top 50 predicted miRNAs related to gastric neoplasms obtained by WINMDA based on known associations in the HMDD database.

Top 1–25 miRNAs	Evidence	Top 26–50 miRNAs	Evidence
hsa-mir-146b	26673617	hsa-mir-20a	29450946
hsa-mir-130a	25834316	hsa-mir-375	21343377
hsa-mir-21	miR2Disease	hsa-mir-17	30024601
hsa-mir-146a	28922434	hsa-mir-222	miR2Disease
hsa-mir-155	26950485	hsa-mir-101	28944848
hsa-mir-145	miR2Disease	hsa-mir-199a	24655788
hsa-mir-143	miR2Disease	hsa-mir-22	28482669
hsa-mir-200a	25740983	hsa-mir-196a	24527072
hsa-mir-200b	25740983	hsa-mir-223	22270966
hsa-mir-126	26464628	hsa-mir-7	26261179
hsa-mir-200c	27766962	hsa-mir-34c	18803879
hsa-let-7a	miR2Disease	hsa-mir-122	29509059
hsa-mir-141	miR2Disease	hsa-mir-218	27696291
hsa-mir-34a	25834316	hsa-mir-34b	unconfirmed
hsa-mir-142	21343377	hsa-mir-10b	25190020
hsa-mir-31	19598010	hsa-mir-103a	29754469
hsa-mir-16	miR2Disease	hsa-mir-27a	miR2Disease
hsa-mir-192	24981590	hsa-mir-150	20067763
hsa-mir-486	26895105	hsa-mir-18a	26950485
hsa-mir-221	miR2Disease	hsa-mir-19a	22802949
hsa-mir-107	miR2Disease	hsa-mir-106a	miR2Disease
hsa-let-7f	21533124	hsa-mir-9	28418879
hsa-let-7g	25972194	hsa-mir-451a	unconfirmed
hsa-mir-133b	23296701	hsa-mir-124	27041578
hsa-mir-125b	24846940	hsa-mir-1	25874496

**Table 6 ijms-20-00110-t006:** The potential top 50 predicted miRNAs related to prostate neoplasms obtained by WINMDA based on known associations in the HMDD database.

Top 1–25 miRNAs	Evidence	Top 26–50 miRNAs	Evidence
hsa-mir-143	dbDEMC and miR2Disease	hsa-mir-15a	dbDEMC and miR2Disease
hsa-mir-182	dbDEMC and miR2Disease	hsa-mir-181b	dbDEMC and miR2Disease
hsa-mir-96	dbDEMC and miR2Disease	hsa-mir-375	dbDEMC and miR2Disease
hsa-mir-34a	dbDEMC and miR2Disease	hsa-mir-200a	dbDEMC
hsa-mir-210	miR2Disease	hsa-mir-34b	dbDEMC
hsa-mir-150	dbDEMC	hsa-mir-34c	dbDEMC
hsa-mir-92a	Unconfirmed	hsa-let-7b	dbDEMC and miR2Disease
hsa-mir-141	miR2Disease	hsa-mir-218	dbDEMC and miR2Disease
hsa-mir-21	dbDEMC and miR2Disease	hsa-mir-101	dbDEMC and miR2Disease
hsa-mir-222	dbDEMC and miR2Disease	hsa-mir-124	dbDEMC
hsa-mir-31	dbDEMC and miR2Disease	hsa-mir-223	dbDEMC and miR2Disease
hsa-mir-146b	25712341	hsa-let-7a	dbDEMC and miR2Disease
hsa-mir-221	dbDEMC and miR2Disease	hsa-mir-224	dbDEMC and miR2Disease
hsa-mir-203	26499781	hsa-mir-205	dbDEMC and miR2Disease
hsa-mir-126	dbDEMC and miR2Disease	hsa-let-7d	dbDEMC and miR2Disease
hsa-mir-200b	Unconfirmed	hsa-mir-1	dbDEMC
hsa-mir-200c	dbDEMC	hsa-let-7c	dbDEMC and miR2Disease
hsa-mir-146a	miR2Disease	hsa-mir-127	dbDEMC and miR2Disease
hsa-mir-17	miR2Disease	hsa-mir-135b	dbDEMC
hsa-mir-100	dbDEMC and miR2Disease	hsa-mir-214	dbDEMC and miR2Disease
hsa-mir-16	dbDEMC and miR2Disease	hsa-mir-93	26124181
hsa-mir-199a	dbDEMC and miR2Disease	hsa-mir-708	22552290
hsa-mir-20a	miR2Disease	hsa-mir-155	dbDEMC
hsa-mir-133b	dbDEMC	hsa-mir-133a	dbDEMC
hsa-mir-27b	dbDEMC and miR2Disease	hsa-mir-195	dbDEMC and miR2Disease

**Table 7 ijms-20-00110-t007:** Effects of *w* on the prediction performance of WINMDA when *T* = 16.

Disease	WINMDA	BNPMDA	PBMDA	WBSMDA	RLSMDA
Breast neoplasms	44	48	46	36	42
Colon neoplasms	47	45	47	45	46
Gastric neoplasms	48	43	46	43	44
Kidney neoplasms	45	43	42	42	45
Liver neoplasms	48	45	45	46	46
Prostate neoplasms	48	44	45	42	44

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
