# Peer review of "A Novel Approach Based on a Weighted Interactive Network to Predict Associations of MiRNAs and Diseases"

_ijms, 2018, doi:10.3390/ijms20010110_

Reviewer 1 Report

In the manuscript entitled “A novel Approach based on Weighted Interactive Network to Predict Association of MiRNAs and Disease” Zhao et al propose a new ingenious method to predict miRNAs for various diseases and to predict diseases for the different miRNAs. I consider the manuscript solid and valuable, but multiple changes are needed before publication.

1.    Abbreviations are used incorrectly and inconsistently throughout the manuscript, e.g. miRNA, ncRNA etc;

2.    Some sentences have no meaning and are incomplete: line 43, regarding miR-381;

3.    The text needs to be corrected for English errors by a native speaker, e.g. lines 67-69;

4.    It should be made more clear what the model brings new compared to other models and how it can be used by researchers in the future;

5.    The material and methods section should be introduced before the results so that the manuscript is easy to read and understand, also the authors refer to some equations in the results as “above” which are actually at the end of the manuscript (line 145);

6.    The manuscript contains multiple repetitive phrases which should be removed;

7.    References are missing: line 169, line 171, line 183;

8.    Line 192: sensitizes – to what? Be more specific. Line 197 be specific about what cancer you are referring to. 

9.    Use statistic methods to prove that your method is superior to already established methods (line 239);

10.  What to the authors propose for the situation in which a miRNA plays an important role in multiple disease? Can the model be used to detect the diseases in which the miRNA is most effective?

11.  Conclusion section contains very repetitive statements, should be rewritten.

Overall I consider the model proposed is interesting, but it must be made clear which are the advantages and how it can be used. Additionally, the paper could benefit from English and stylistic refinements. 

Author Response

Response to Reviewer 1 Comments

 Point 1: Abbreviations are used incorrectly and inconsistently throughout the manuscript, e.g. miRNA, ncRNA etc;

Response 1: Thank you very much to point out the abbreviations issues in our manuscript. According to the comments from you and the editors, we modified the content of the abbreviation part of the manuscript., conscientiously.

Point 2: Some sentences have no meaning and are incomplete: line 43, regarding miR-381;

Response 2: Thank you very much to point out the sentence structure and grammatical issues in our manuscript. According to the comments from you and the editors, we made a correction in the revised version of the manuscript. Line 43, the statements of ‘and the miR-424 and miR-381 have been demonstrated to MiR-424 and miR-381 play an important role in tumor regulation, expression and even treatment.’ were corrected as ‘MiRNA expression microarray analysis showed that miR-145 and miR-1 expression are significantly down regulated in colon cancer tissues [9], and the miR-424 and miR-381 play important roles in tumor regulation, expression and even treatment [10].’.

Point 3: The text needs to be corrected for English errors by a native speaker, e.g. lines 67-69;

Response 3: Thank you very much to point out the sentence structure and grammatical issues in our manuscript. According to the comments from you and the editors, we made a correction in the revised version of the manuscript. Line 67-69, the statements of ‘In all these above mentioned computational models, known miRNA-disease associations were represented as a adjacency matrix A, in which, each element will be 0 or 1, i.e., if the association between the ith-miRNA and the jth-disease is known, then there is A(i,j)=1, and otherwise, then there is A(i,j)=0.’ were corrected as ’ In most of the above mentioned computational models, Researchers constructed an adjacency matrix A to represented relationships between miRNA and disease. Specifically, if association term between miRNA mi and disease dj is recorded in database, the matrix A(i,j)=1, otherwise 0.’.

Point 4: It should be made more clear what the model brings new compared to other models and how it can be used by researchers in the future;

Response 4: Comparing with several state-of-the-art computational models, the strong point of WINMDA lies in the construction of weighted interactive network and the introduction of shortest path between nodes in the weighted interactive network. This also means that WINMDA proposes an idea to improve the sparseness of the adjacency matrix A and doesn’t need negative samples to predict potential miRNA-disease associations simultaneously. All prediction results of potential miRNA-disease association have been shown in Supplementary File 1, researchers could use it to guide biological experiments in the future.

Point 5: The material and methods section should be introduced before the results so that the manuscript is easy to read and understand, also the authors refer to some equations in the results as “above” which are actually at the end of the manuscript (line 145);

Response 5: We are sorry for our incorrect composition, we typeset our manuscript according to the format given by the journal. And ‘above’ was deleted.

Point 6: The manuscript contains multiple repetitive phrases which should be removed;

Response 6: Thank you very much to point out the multiple repetitive phrases issues in our manuscript, we have removed repetitive phrases in a new version of our manuscript.

Point 7: References are missing: line 169, line 171, line 183;

Response 7: Thank you very much to point out the References issues, we have added some references in a new version of our manuscript.

Point 8: Line 192: sensitizes – to what? Be more specific. Line 197 be specific about what cancer you are referring to.

Response 8: According to the comments from you and the editors, Line 192, the statement of ‘some researchers have confirmed that the overexpression of miR-143 (Ranked 1st in WINMDA forecast list) will reduce cell proliferation and migration of mutant KRAS HCT116 colon cancer cells, and sensitize both mutant KRAS (HCT116 and SW480) [28].’ were corrected as ‘some researchers have confirmed that the overexpression of miR-143 (Ranked 1st in WINMDA forecast list) will reduce cell proliferation and migration of mutant KRAS HCT116 colon cancer cells [32].’, Line 197, ‘colon cacner’ was added.

Point 9: Use statistic methods to prove that your method is superior to already established methods (line 239);

Response 9: We are sorry that we did not clearly understand the meaning of the above comment. In a new version of our manuscript, we evaluated the predictive power of five methods for six important diseases. We counted the top 50 potential disease-associated miRNAs predicted by the five methods, and recorded how many of the potential associations were confirmed by other databases and literature. The results have been shown in table 7.

Point 10: What to the authors propose for the situation in which a miRNA plays an important role in multiple disease? Can the model be used to detect the diseases in which the miRNA is most effective?

Response 10: MiRNA plays an important role in multiple disease, therefore, it is necessary to study the association between miRNAs and diseases in depth and explore the potential relationship between miRNA and some human diseases. Our method scores the association of each miRNA-disease, and the higher the score, the more relevant the microRNA is to the disease. Therefore, the most relevant miRNAs for each disease can be found.

Point 11: Conclusion section contains very repetitive statements, should be rewritten. 

Response 11: According to the comments from you and the editors, we rewrote the discussion section and conclusion section.

Discussion

Increasing studies based on biological experiments have indicated that miRNAs were closely related to the occurrence of many diseases that are seriously harmful to human health, and the identification of potential miRNA-disease associations can not only play an important role in the diagnosis, treatment, and prevention of disease, but also effectively addresses the high cost and long-term shortcomings of traditional biological experiments. In this article, we developed a novel prediction model called WINMDA was to predict potential relationships between miRNAs and diseases based on premises that functionally similar miRNAs may regulate similar diseases and similar diseases tend to associate functionally similar miRNAs. And in WINMDA, we integrated disease semantic similarity, Gaussian interaction profile kernel similarity and miRNA function similarity firstly, and then constructed a weighted interactive network for potential miRNA-disease prediction. The important difference between WINMDA and previous state-of-the-art prediction models is that the problem of limited known miRNA-disease associations is considered in WINMDA and the shortest paths in the weighted interactive network are adopted to solve the problem. Moreover, we evaluate the predictive performance of WINMDA through LOOCV (including globe LOOCV and local LOOCV), K-fold cross validation and several case studies. Experimental results show that WINMDA can uncover potential disease-miRNA candidates effective, which means that it can be used as a reliable and accurate Calculation model for finding potential miRNA-disease associations effectively.

Although WINMDA achieved effective performance in predicting candidate relationships between miRNAs and diseases, there are still some existing limitations that can be improved in the future. Firstly, the parameter T and w play important roles in WINMDA, and the selection of suitable values for T and w are critical problems that shall be addressed in future studies. Secondly, the assigned weight may not be accurate enough, as it was on the basis of premises that functionally similar miRNAs may regulate similar diseases and similar diseases tend to associate functionally similar miRNAs. Finally, a weighted interactive network was constructed in WINMDA based on the disease similarity, miRNA similarity and known miRNA-disease associations, the performance of WINMDA will be further improved once considering more databases storing other information about diseases, miRNAs and miRNA-disease associations.

Conclusion

In this article, the effective predictive performance of WINMDA is mainly due to the following reasons: (1) The sematic disease similarity, functionally miRNA similarity and Gaussian interaction profile kernel similarity were integrated. (2) We proposed a new method for calculating the semantic similarity of diseases. (3) We constructed a weighted interactive network based disease similarity, miRNA similarity and known miRNA-disease associations. (4) The concept of T most similar neighbours was introduced. (5) A algorithm for searching the shortest path in the weighted interactive network was introduced. (6) In the future work, multiple heterogeneous biological data can be collected and pre-processed to be utilized in the weighted interactive network, thus improving performance of prediction algorithms.

Reviewer 2 Report

I read the work very carefully. the manuscript is certainly interesting and deals with a topic very important in studying miRNA and diseases association.

And I think that it could be important for the journal.

However, as I already clarified, I don't fell qualified to judge the paper from a  bioinformatic point of view.

So I strongly suggest that a bioinformatic expert read the manuscript

Author Response

Point 1: I read the work very carefully. the manuscript is certainly interesting and deals with a topic very important in studying miRNA and diseases association.

And I think that it could be important for the journal. However, as I already clarified, I don't fell qualified to judge the paper from a  bioinformatic point of view.

So I strongly suggest that a bioinformatic expert read the manuscript.

Response 1: Thank you very much for your comments and suggestions.

Reviewer 3 Report

Zhao et al. developed a network-based approach to predict potential association between diseases and miRNAs. miRNAs play important roles in disease development, so it is critical to find candidate miRNAs resulting into certain diseases or serving as drug targets. Their approach outperforms other similar algorithms such PBMDA, WBSMDA, RLSMDA. However, the improvement seems not very impressive with AUC gain ~0.008 (global) and ~0.03 (local) compared to second best approach. The authors also didn’t compare their algorithm to another very similar algorithm BNPMDA which is a network-based approach as well. Overall, the manuscript is technically sound, but many parts of this approach is not very novel. In addition, the writing needs to be significantly improved given there are quite a few spelling and grammar errors.

1)      The authors should compare their approach to BNPMDA which was published early 2018. It seems BNPMDA also uses the similarity between miRNAs/diseases to find the miRNA-disease association.

2)      Initially, the authors use the term “interactive network”, but all of sudden they switched to “interaction network’ in a few places. Please use the right terminology and be consistent. Also, the full name of WINMDA needs to be written out at the first place.

3)      Need to discuss several potential miRNA-diseases examples to show why they are associated. I bet there should be some known miRNAs associated with those 6 neoplasms. So how many of them were recovered among top pairs?

4)      In figure 5, what is the optimal number of k?

5)      The code should be publicly available. So other groups can reproduce the results and apply it to their own studies.

Author Response

Point 1: The authors should compare their approach to BNPMDA which was published early 2018. It seems BNPMDA also uses the similarity between miRNAs/diseases to find the miRNA-disease association.

Response 1: In the revised version of our article, according to above comments, we added some comparative experiments.

Point 2: Initially, the authors use the term “interactive network”, but all of sudden they switched to “interaction network’ in a few places. Please use the right terminology and be consistent. Also, the full name of WINMDA needs to be written out at the first place.

Response 2: Thank you very much to point out the sentence structure and grammatical issues in our manuscript. According to the comments from you and the editors, we made a correction in the revised version of the manuscript. ‘interaction network’ were corrected as ‘interactive network’.

Point 3: Need to discuss several potential miRNA-diseases examples to show why they are associated. I bet there should be some known miRNAs associated with those 6 neoplasms. So how many of them were recovered among top pairs?

Response 3: We have discussed some of the specific reasons for the potential miRNA-diseases association in the Case studies section. The result of the leave-one-out cross-validation we performed is based on the ranking of the known miRNA-disease associations predicted scores in all predicted scores. According to the results, we can see that the predicted scores of most known associations are very high, there are many known associations in top pairs.

Point 4: In figure 5, what is the optimal number of k?

Response 4: Thank you very much to point out the figure issue, we have modified ‘figure 5’ in the new version of our manuscript.

Point 5:   The code should be publicly available. So other groups can reproduce the results and apply it to their own studies.

Response 5: All prediction results of potential miRNA-disease association have been shown in Supplementary File 1, researchers could use it to guide biological experiments in the future.

Round  2

Reviewer 3 Report

The authors have addressed all my comments and I believe the revised manuscript is much improved

Author Response

Point 1: The authors have addressed all my comments and I believe the revised manuscript is much improved.

Response 1: Thank you very much.